# Diversity, composition, and networking of saliva microbiota distinguish the severity of COVID-19 episodes as revealed by an analysis of 16S rRNA variable V1-V3 region sequences

Violeta Larios Serrato,[1] Beatriz Meza,[2,3,4] Carolina Gonzalez-Torres,[5] Javier Gaytan-Cervantes,[5] Joaquín González Ibarra,[6] Clara Esperanza Santacruz Tinoco,[7] Yu-Mei Anguiano Hernández,[7] Bernardo Martínez Miguel,[7] Allison Cázarez Cortazar,[7] Brenda Sarquiz Martínez,[7] Julio Elias Alvarado Yaah,[7] Antonina Reyna Mendoza Pérez,[8] Juan José Palma Herrera,[9] Leticia Margarita García Soto,[9] Adriana Inés Chávez Rojas,[10] Guillermo Bravo Mateos,[11] Gabriel Samano Marquez,[11] Concepción Grajales Muñiz,[12] Javier Torres[4]

**ABSTRACT**   Studies on the role of the oral microbiome in SARS-CoV-2 infection and severity of the disease are limited. We aimed to characterize the bacterial communities present in the saliva of patients with varied COVID-19 severity to learn if there are differences in the characteristics of the microbiome among the clinical groups. We included 31 asymptomatic subjects with no previous COVID-19 infection or vaccination; 176 patients with mild respiratory symptoms, positive or negative for SARS-CoV-2 infection; 57 patients that required hospitalization because of severe COVID-19 with oxygen saturation below 92%, and 18 fatal cases of COVID-19. Saliva samples collected before any treatment were tested for SARS-CoV-2 by PCR. Oral microbiota in saliva was studied by amplification and sequencing of the V1-V3 variable regions of 16S gene using an Illumina MiSeq platform. We found significant changes in diversity, composition, and networking in saliva microbiota of patients with COVID-19, as well as patterns associated with severity of disease. The presence or abundance of several commensal species and opportunistic pathogens were associated with each clinical stage. Patterns of networking were also found associated with severity of disease: a highly regulated bacterial community (normonetting) was found in healthy people whereas poorly regulated populations (disnetting) were characteristic of severe cases. Characterization of microbiota in saliva may offer important clues in the pathogenesis of COVID-19 and may also identify potential markers for prognosis in the severity of the disease.

**IMPORTANCE**   SARS-CoV-2 infection is the most severe pandemic of humankind in the last hundred years. The outcome of the infection ranges from asymptomatic or mild to severe and even fatal cases, but reasons for this remain unknown. Microbes normally colonizing the respiratory tract form communities that may mitigate the transmission, symptoms, and severity of viral infections, but very little is known on the role of these microbial communities in the severity of COVID-19. We aimed to characterize the bacterial communities in saliva of patients with different severity of COVID-19 disease, from mild to fatal cases. Our results revealed clear differences in the composition and in the nature of interactions (networking) of the bacterial species present in the different clinical groups and show community-patterns associated with disease severity. Characterization of the microbial communities in saliva may offer important clues to learn ways COVID-19 patients may suffer from different disease severities.

**KEYWORDS**   COVID-19, microbiota, microbial networks, disease severity, viral infections

Address correspondence to Javier Torres, uimeip@gmail.com.

The authors declare no conflict of interest.

See the funding table on p. 18.

SARS-CoV-2 infection is the most severe human pandemic in the last 100 years, with over 599 million cases and 6.46 million deaths worldwide as of August 2022. By 2.5 years after it first appeared, the virus has become endemic, with occasional outbreaks that have sometimes required re-implementation of public health containment measures. SARS-CoV-2 infects human cells expressing the angiotensin-converting enzyme 2 (ACE2), which is used as receptor for the spike protein. This property enables the virus to infect different cells in different organs, causing a multisystem, multiorgan infection (1, 2). The outcome of infection ranges from asymptomatic or mild episodes to severe and fatal cases (3). Severity of disease is known to be determined by multiple factors, including host and viral characteristics.

Another factor that may determine severity of disease is the human microbiome, which is known to play a major role in the modulation of other infections. A healthy microbial community may mitigate the transmission, symptoms, and severity of viral infections (4). Members of our normal bacterial communities may also inhibit viral replication and modulate the inflammatory response to counteract the infection and prevent immune mediated tissue damage (4). However, studies on the role of the microbiome in SARS-CoV-2 infection and on severity of the disease have been limited. All studies reported to date have found that patients infected with SARS-CoV-2 have an altered microbiome composition in nasopharyngeal samples, with reduced diversity, although differences in bacterial composition vary among studies. One study reported that *Propionibacteriaceae* were significantly more abundant and *Corynebacterium accolens* significantly decreased in infected patients (5). Others reported that *Prevotella* and *Alloprevotella* were increased in abundance in severe cases (6). Metatranscriptomic analyses of nasopharyngeal swabs and sputum samples also found a reduced diversity in patients with COVID-19 pneumonia when compared with non-COVID-19 pneumonia. Different species of *Prevotella*, *Veillonella*, *Haemophilus*, *Fusobacterium,* and *Gemella* showed a reduced abundance in the COVID-19 cases (7). The oral mucosa has been recognized as an important site for SARS-CoV-2 infection and as a source for spreading the infection to the upper and lower respiratory tract (8). It has been reported that the oral microbiome (tongue swabs) forms a dysbiotic pattern in long-COVID cases, with higher abundance of microbiota that induce inflammation, including *Prevotella* and *Veillonella* (9). Diversity of oral (tongue scraping) microbiome was also found reduced in patients with COVID-19, with increases in bacteria producing lipopolysaccharide and decreases in bacteria producing butyric acid (10).

The oral cavity hosts over 1,000 bacterial species, representing the second largest and most diverse microbial community in the human body after the gut (11). The oral microbiota has been shaped as a result of coevolution with the human mouth. Local niches in the mouth supply shelter and nutrients, whereas the microorganism complement the physiological needs of the host, including nutrients, immune training, and host defense (12). The oral microbiota is formed by a collection of compositionally distinct communities that reflect the array of diverse microenvironments present in the different regions of the mouth. These communities usually grow as highly structured symbiotic biofilms, linked through physical and metabolic associations that confer a fitness advantage to the entire microbial community and make them particularly stable and resilient to microenvironmental changes (13). The salivary microbiota has been shown to be a conglomerate of bacteria shed from oral surfaces in the pharynx, the tongue and the tonsils as the main sites of origin (14). Thus, saliva is an appropriate sample that mirrors the microbiota from different regions of the oral cavity. Saliva can also be aspirated and reach the lungs, representing an important source of infection for the respiratory tract. Indeed, studies have suggested a relationship between oral hygiene and respiratory diseases, including asthma, chronic obstructive pulmonary disease, and pneumonia (15).

In the present work, we aimed to characterize the bacterial communities present in the saliva of patients with COVID-19. The study included groups of patients with different severity of disease, from mild to fatal cases, to determine if there were differences in the

composition of the microbiome among clinical groups. The results demonstrated clear differences in diversity, composition and networking of the bacterial species present in the saliva of the different clinical groups.

## MATERIALS AND METHODS

### Patients studied

The study was done during the period of June 2020 to January 2021, a period of high epidemiological uncertainty and strict containment measures because of COVID-19. Therefore, recruitment of patients was done by the attending health personnel. Under these circumstances, we did not have enough information to estimate a sample size, nor were we certain of the feasibility to reach an expected number of cases. Thus, patients were invited to the study as they were presented with suspicious symptoms, and before a confirmed diagnosis in mild cases, or within 5 days of hospitalization in severe confirmed cases. Final sample size of the clinical groups was based on the inclusion-exclusion criteria. Patients willing to participate were asked to sign an informed letter. The study was approved by the IRB Comisión Nacional de Investigación, Instituto Mexicano del Seguro Social, Mexico (registry number R-2020-785-053).

Asymptomatic cases (AC) were individuals without symptoms, no previous COVID-19 infection (as referred by the patient) or SARS-CoV-2 vaccine, and no antibiotics in the last 4 weeks. Smokers and those with any chronic disease were excluded. Mild cases were ambulatory patients with mild respiratory symptoms (fever, cough, headache, odyno-phagia, myalgias) presenting for COVID-19 diagnosis. They were sampled before any treatment, including antibiotics, and followed until recovery. After testing for SARS-CoV-2 by PCR test (COBAS 6800, Merck México, Mexico City), they were classified as ambulatory (mild cases) positive (AP) or negative (AN) for the infection. Patients who during follow up required hospitalization because they developed severe symptoms were excluded from this group and included in the hospitalized group. Severe patients were cases that required hospitalization because of severe symptoms (HP), particularly with an oxygen saturation below 92% and the presence of comorbidities of risk including hypertension, diabetes, morbid obesity, immunocompromise, cardiovascular or neurological diseases, chronic renal failure, tuberculosis or neoplasia. Patients were usually hospitalized within the first 7 days after symptoms started and followed until discharged because of recovery or improvement (treated at home until recovery) or because of death. For the analyses, patients who died were included in a group of deceased cases (DHP).

### Saliva samples

A volume of 10 mL of saline solution (0.85% NaCl) was given to patients, who were asked to thoroughly wash the mouth and spit back into a 50 mL plastic tube. Samples were immediately transported to a central laboratory for SARS-CoV-2 diagnosis using a PCR test to amplify a fraction of the spike and N protein genes. On arrival, samples were immediately inactivated by heating at 65°C for 30 min. After the diagnosis, samples were sent to our laboratory for microbiome studies. Transport of the samples was done following international regulations for safety, including special multiple packing with dry ice. Once received in our lab, samples were frozen to −70°C until studied. All five groups of patients were recruited during the same period of time.

### DNA extraction

Saliva samples (1 mL) were centrifuged for 10 min at 5,000 × g, then bacterial pellets were suspended and incubated at 37°C for 3 h with 180 µL of enzyme solution (20 mg/mL lysozyme; 20 Mm Tris-HCl, pH 8.0; 2 mM EDTA; Triton 1.2%). Subsequently, DNA was extracted using the QIAamp DNA mini kit (Qiagen) according to the manufacturer's protocol.

## Preparation of the amplicon libraries from the 16S rRNA hypervariable regions V1-V3 and sequencing

The amplicons of the V1-V3 hypervariable regions of the 16S rRNA gene were generated using previously published primers (16) (Table S1), which were ligated to the adapter sequences (17) (Table S2) and used to assemble the DNA libraries. For library assembly, 25 ng of DNA were mixed with 12.5 µL of Go-Taq green master mix enzyme (Promega) and 10 µM of each primer (Table S1), and amplified using the following conditions: 3 min at 98° followed by 25 cycles (20 s at 98°C, 30 s at 65°C, 30 s at 70°C), 5 min at 72°C and 4°C hold. Libraries were normalized and sequenced using the 2 × 251 cycle configuration, with 20% Phix control, and 100 µM of the sequencing primers and placed in positions 12, 13, and 14 of the sequencing cartridge. Sequencing of the libraries was performed on the MiSeq platform (Illumina, San Diego California, USA).

## Bioinformatics analysis

### Quality control and taxonomic classification

Paired sequencing fastq files were QC inspected for Phred values and for absence of adapters using FastQC v0.11.9 (18). The data were processed using the DADA2 v1.20.0 pipeline (19). Standard filter parameters (maxN = 0, truncQ = 8, and maxEE = 2, lengths below 200 bp were discarded) were used. The process of reads continued with dereplication filtering, and removal of chimera formation (representing < 2%) using the removeBimeraDenovo option. 16S sequences associated with chloroplast or mitochondria were removed. The sequences were grouped into Amplicon Sequence Variants (ASVs) with the naive RDP Bayesian classifier of DADA2, and taxonomic classification was assigned to the species level using the expanded Human Oral Microbiome Database 16S V1-V3 training set, eHOMD v15.1 (20) (http://www.homd.org, consulted 5 November 2021).

## Alpha and beta diversity

Normalization geometric mean was calculated for each ASVs across all samples using Total Sum Scaling (TSS) (21) with MicrobiomeMarker v1.3.3 package (22). The alpha and beta diversity were determined using the Phyloseq v1.42.0 (23), Vegan v2.6-4 (24), and ggplot2 v3.4.1 (25) packages with a prevalence of 10%, to obtain the Chao1 (richness estimator), Fisher (abundance estimator), and the Shannon and Simpson (diversity and evenness estimators) values. Stacked bar graphs were created to observe the bacterial composition for each group, with a minimal prevalence of 0.1 for species and family. The ape package v5.3 (26) was used for the generation of the phylogenetic tree. Statistical significance was evaluated with a univariate ANOVA analysis and $P$-values ≤ 0.05 were considered significant. Between-group differences in beta diversity were assessed using principal coordinate analysis (PCoA) with unweighted and weighted UniFrac to visualize differences in COVID-19-associated bacterial communities. Significance with Bray–Curtis dissimilarity index was assessed by calculating non-parametric permutational multivariate analysis of variance (PERMANOVA) with 10,000 permutations, using the adonis and beta disper functions. Analysis of similarities (ANOSIM) (27) and homogeneity of group dispersions (PERMDISP) were also performed.

## Dysbiosis

The ASVs and taxa generated by DADA2 were used to build a Phyloseq object. A dysbiosis score was calculated using dysbiosisR v1.0.4 package, the selection measure was median variation (28) and as reference the AC group and Bray–Curtis dissimilarity matrix. Additionally, a graph of receiver operating characteristic (ROC) and area under the curve (AUC) values was calculated with pROC v1.18.0 package.

## Biomarker discovery

The differences in taxa between the experimental groups were studied in order to identify the ASVs that significantly distinguished each group (marker bacteria). The analyses were done using different models, including Random Forest (RF) using Caret v6.0-94 (29) and MLeval v0.3 package (30) with 1,000 trees to build the model, and MicrobiomeAnalyst v2.0 (31). For cross-validation values were multiplied by 10, and the training set was 90% while the test set was 10%. In addition, for further validation, a ROC plot and AUC table were determined. Analyses also included a LEfSe test using microbiomeMarker v1.3.3 package (22) with a q-value $\leq$ 0.1 and linear discriminant analysis (LDA score $\geq$ 2.0). To analyze the differences in taxa abundance among groups, the Fold Change (FC) and the FDR-adjusted were calculated using DESeq2 (32), and for normalization geometric mean was calculated for each ASVs across all samples using TSS. The data were previously filtered based on the significant results of Random Forest and LefSe. Volcano plots were constructed with EnhancedVolcano v1.12.0 (33).

## Composition of the core microbiome

The microbiome v1.17.3 package with the core function (34) was used to calculate taxa present in most samples among all clinical groups. Samples were filtered applying a prevalence of 0.5 and a relative abundance of 0.20 (threshold for absence/presence) among the samples. The visualization of the data were carried out through a heatmap with the plot core function, the variables captured were relative abundance and the ASVs of the output samples for all the studied groups.

## Determination of co-abundance networks

A single association, co-occurrence network was built with the NetCoMi v1.0.2 in R, with the sparse correlations for compositional data (SparCC) method (35). The resulting correlation matrix was utilized in network models to define links between taxa, if the absolute pairwise correlation between two taxa was greater than 0.25. A t-test was applied (alpha = 0.001) to reduce the network to a tractable size with a false discovery adjustment to select edges to include in the network (sparsification). Network features, including degree, betweenness, closeness centrality, and modularity computation enabled identification of hubs (quantile set at 0.9). The network, including the subclusters was constructed based on the fast, greedy algorithm, applying agglomeration at the species level and using the netConstruct function (measure of correlation of "pearson", zeroMethod of "multRepl," and normMethod "clr") and the netAnalyze function (clust_fast_greedy method). Graphics were generated using the Fruchterman–Reingold layout algorithm of igrap. The size of the nodes was adjusted by a normalization of the counts, a color was assigned to each subcluster within the net, and the single nodes were removed. Estimated associations are shown with green connections for positive or red for negative correlations.

## Metabolic routes

The inference of metabolic pathways were predicted with KEGG Orthology (KO) Level three information (36) and Phylogenetic Investigation of Communities by Reconstruction of Unobserved States 2 (Picrust2 v.2.1.3-b) (37), which predicts gene family abundance. To obtain relative abundance pathways, we used the scaled (TSS method), the ASV table and representative sequences, with the default options in picrust2_pipeline.py. The z-score of relative abundances and clustering were calculated using the package pheatmap v1.012. A linear discriminant analysis (LDA) integrated with effect size (LEfSe) of the relative abundance of KEGG pathways was performed. Microbial v.0.0.20 package with the ldamarker function and a $P$-value of 0.05 was used and the TMM normalization method was selected. To build the graphs, the plotLDA function was used with padj values of 0.05 and LDA $\geq$ 5.

## RESULTS

The number of patients that consented to participate was close to 500 and from those we were able to have enough saliva sample and follow up clinical data in 390 cases. After excluding patients who did not fulfill inclusion criteria, including quality and amount of sample, 314 saliva samples were sequenced. In the end, 282 samples (88.5%) passed the Q30 quality value with an average of 200,000 reads per sample and a total yield of 16.1 Gb, distributed as described in Table 1.

### Diversity and abundance of bacterial species differ among patients with different severity of COVID-19

Microbial diversity indexes of the saliva samples are presented in Fig. 1A; results show that richness (Chao index) was higher in the asymptomatic individuals but gradually and significantly decreased in the ambulatory, hospitalized, and deceased patients, a result that was further supported by the Fisher analysis. Shannon index however showed an increased value in all symptomatic groups, suggesting an increase in evenness in these patients. Diversity in the microbial composition among groups was analyzed by a PCoA UniFrac and Weighted UniFrac analyses (Fig. 1B), which showed separation of the bacterial communities in the clinical groups, hospitalized severe cases (HP and DHP) clustered significantly apart from mild ambulatory cases and from the asymptomatic group (see P-values in Fig. 1B). The two hospitalized groups, HP and DHP, showed no significant separation. Differences in bacterial structure (according to species) among the groups were significant as the PERMDISP analyses show (Fig. 1B). The degree of dysbiosis in disease groups as compared with the asymptomatic group is shown in Fig. S4, the degree of dysbiosis significantly distinguished asymptomatic people from the disease groups as illustrated by the AUC analysis.

### Pairwise comparisons showed marked and significant differences in bacterial composition among groups

Pairwise differences between groups were studied with the enhanced volcano test (Fig. 2). By amplifying the V1-V3 regions of the 16S rRNA gene and by using the eHOMD database to annotate, we were able to identify most of the AVS to the level of species (20). Compared with the asymptomatic group, *Actinomyces odontolyticus*, *Streptococcus parasanguinis*, *Oribacterium sinus*, *Atopobium parvulum,* and *Streptococcus mutans* (among others) were significantly more associated with the mild ambulatory group (Fig. 2 AP vs AC). In contrast, *Prevotella intermedia*, *Porphyromonas gingivalis*, *Alloprevotella* sp., and *Prevotella oris* were more associated with healthy adults. When the group of hospitalized patients was compared with the healthy individuals, *Leptotrichia* sp., *Escherichia coli*, *Staphylococcus epidermidis,* and *Prevotella oris* were significantly more associated with HP patients (Fig. 2 HP vs AC). *Haemophilus* sp. HTM259, *Porphyromonas gingivalis*, *Actinomyces* sp. HTM169, *Haemophilus parainfluenzae,* and others were more associated with the asymptomatic group. *Acinetobacter baumannii*, *Capnocytophaga granulosa*, *Prevotella melaninogenica*, *Granulicatella adiacens*, *Prevotella salivae*, and *Veillonella parvula* were significantly more associated with the DHP fatal patients compared to asymptomatic patients (Fig. 2 DHP vs AC). In contrast, *Granulicatella elegans*,

**TABLE 1** Characteristics of patients with different severity of COVID-19, studied for oral microbiome in saliva samples

| Group of patients | No. studied | Age mean ± SD | Sex ratio male:female |
|---|---|---|---|
| Asymptomatic | 31 | 27.8 ± 7.7 | 0.78 |
| Mild negative[a] | 73 | 38.2 ± 12.1 | 0.59 |
| Mild positive[a] | 103 | 41 ± 13.1 | 0.77 |
| Severe positive[b] | 57 | 50.8 ± 13.9 | 1.94 |
| Deceased[b] | 18 | 74.4 ± 6.5 | 3.3 |

[a]Ambulatory patients.
[b]Hospitalized patients.

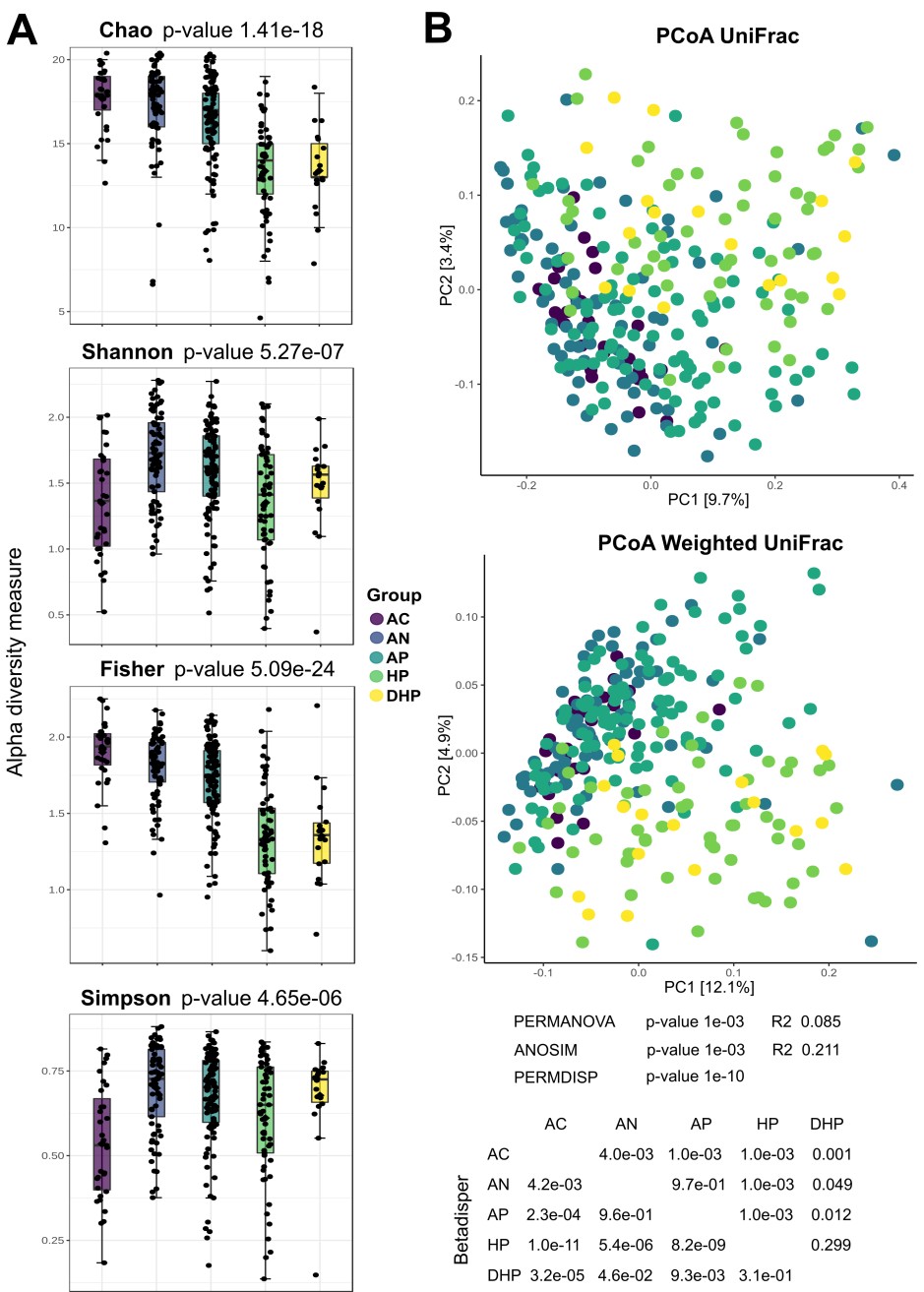

**FIG 1** Indices of microbial diversity of bacterial species in saliva of patients with different severity of COVID-19 disease. (A) Chao, Shannon, Fisher, and Simpson index; (B) Beta diversity with the PCoA UniFrac and Weighted UniFrac analyses. Changes in community structure and in beta diversity were evaluated with permutational multivariate analysis of variance (PERMANOVA), analysis of similarities (ANOSIM) and permutational analysis of multivariate dispersions (PERMDISP) and showed significant differences ($P < 0.001$). Permutation test for homogeneity of multivariate dispersions (Betadisper) was also significant ($P < 0.05$) for most pairwise comparisons, except for HP vs DHP.

*Haemophilus parainfluenzae*, *Veillonella* sp. HMT780, *Alloprevotella* sp. HMT308, and *Prevotella intermedia* were more associated with asymptomatic adults. Of interest, when the two groups of hospitalized patients were contrasted, *Acinetobacter baumannii* and *Prevotella salivae* were more associated with deceased patients, whereas *Escherichia coli*, *Leptotrichia* HMT21,5 and *Staphylococcus epidermidis* with the severe HP patients (Fig. 2 DHP vs HP). Thus, *Acinetobacter baumannii* and *Prevotella salivae* were marker species

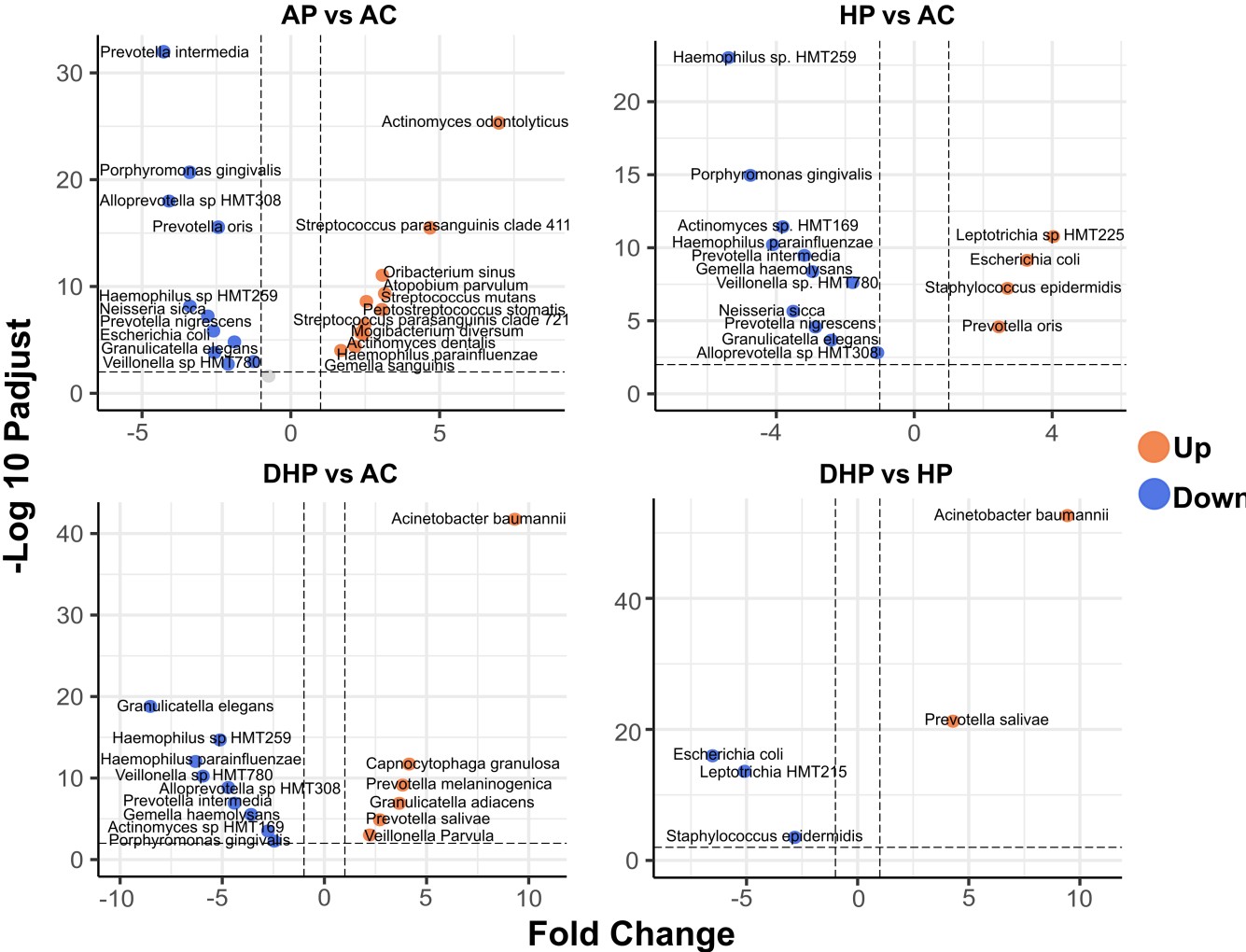

**FIG 2** Differential abundance analysis. Volcano plot shows ASV fold changes (FC) on x-axis and the negative logarithm base 10 of the False Discovery Rate (FDR) on y-axis. Dashed lines reflect threshold, FC ≥ 1.0 (either positive or negative) and FDR > -log (0.05). AP vs AC) comparison between the ambulatory SARS-CoV-2 positive group and the asymptomatic control group; HP vs AC) comparison between the hospitalized SARS-CoV-2 positive group and the asymptomatic control group; DHP vs AC) comparison between the deceased SARS-CoV-2 positive group and the asymptomatic control group; DHP vs HP) comparison between the deceased and the hospitalized group. Species in orange dots presented increased abundance and those in blue decreased abundance.

that differentiated deceased patients from severe hospitalized patients and from asymptomatic individuals.

We also studied a group of patients with mild respiratory disease that were negative for SARS-CoV-2 infection (AN). Compared with the asymptomatic adults (Fig. S1 AN vs AC), these patients had higher abundance of *Actinomyces graevenitzii*, *Streptococcus mutans*, *Peptostreptococcaceae* XI G1, *Actinomyces dentalis,* and *Stomatobaculum longum* ,whereas *Prevotella intermedia*, *Veillonella* sp. HMT780, *Alloprevotella* sp. HMT308, *Escherichia coli,* and *Neisseria sicca* were more abundant in the healthy group. Of note, when the two groups with mild disease (AP and AN) were compared, *Streptococcus parasanguinis* was more abundant in the SARS-CoV-2 patients, whereas *Veillonella dispar*, *Peptostreptococcaceae* X1 G1, *Porphyromonas pasteri*, *Actinomyces dentalis,* and *Actinomyces graevenitzii* were significantly more abundant among the mild non-infected patients (Fig. S1 AP vs AN).

## An all-vs-all comparison revealed species that distinguishes each clinical group

We next examined the differences among all groups by means of the random forest and LEfSe analyses (Fig. 3). In this all-vs-all groups' analyses, random forest (Fig. 3A) found *Haemophilus parainfluenzae*, *Prevotella nigrescens*, *Neisseria sicca*, *Gemella haemolysans,* and *Rothia dentocariosa* among those most significantly distinguishing the asymptomatic group, whereas *Streptococcus parasanguinis*, *Oribacterium sinus*, *Atopobium parvulum,* and *Actinomyces odontolyticus* differentiated the mild ambulatory cases. *Escherichia coli*, *Leptotrichia* sp. HTM225, and *Staphylococcus epidermidis*, distinguished the hospitalized patients, whereas *Veillonella parvula*, *Prevotella melaninogenica*, *Acinetobacter baumannii*, *Granulicatella adiacens*, *Actinomyces dentalis*, *Capnocytophaga granulosa*, *Leptotrichia wadei,* and *Veillonella dispar* strongly differentiated the deceased patients. The model was further validated with an AUC analysis as shown in Fig. S2, where AUC values and the confusion matrix show an excellent differentiation of the AC, AP and HP groups (asymptomatic, moderate, and severe). Behavior of the AN patients suggests its microbiota composition overlaps with the AP group; these two groups are clinically similar but differ in the detection of SARS-CoV-2 infection.

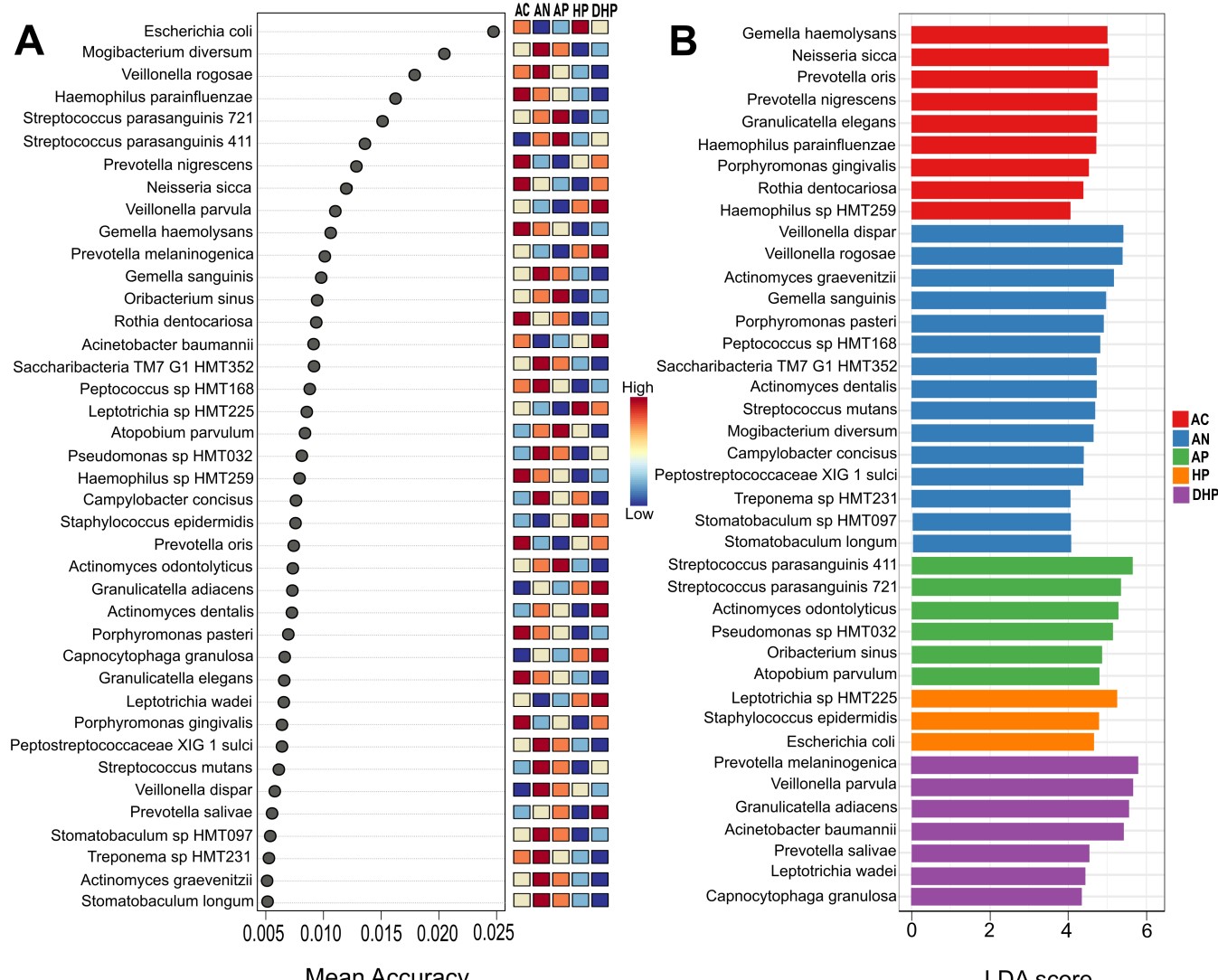

**FIG 3** Biomarker discovery analysis. Study of all-vs-all clinical groups by (A) Random forest and (B) LEfSe test. Groups are indicated to the right of the figures, AC, asymptomatic; AN, ambulatory SARS-CoV-2 negative; AP, ambulatory SARS-CoV-2 positive; HP, hospitalized; DHP, deceased patients.

Results with the linear discriminant analysis (LDA) (Fig. 3B) showed a strong agreement with random forest (Fig. 3A) and of note, the two models pointed to *Prevotella melaninogenica*, *Veillonella parvula,* and *Acinetobacter baumannii* as species strongly distinguishing the group of deceased patients, whereas *Neisseria sicca*, *Haemophilus parainfluenzae,* and *Prevotella nigrescens* were markers for asymptomatic adults.

## A core microbiome analysis show species present in all clinical groups

Finally, we determined the core microbiome (Fig. 4) to learn which species were present in all clinical groups, probably because they are more resilient to changes in the microenvironment. Of note, *Streptococcus pneumoniae* was found present in all clinical groups with a relative abundance of over 10% in over 60% of the patients (see also Fig. S3), highlighting its endurance to changes in the microenvironment regardless of the clinical condition of the patients. *Granulicatella adiacens*, *Veillonella dispar*, *Streptococcus parasanguinis*, *Prevotella melaninogenica,* and *Veillonella parvula* showed a relative abundance of over 1.0% in at least 50% of all patients (Fig. 4; Fig. S3). Other species of *Actinomyces*, *Prevotella,* and *Veillonella* were also among the species in the core microbiome.

Based on all above results we asked how much normobiosis was altered in these patients and estimated a dysbiosis score (Fig. S4A ) and found that the scores significantly differentiated microbiota of mild and severe patients from microbiota of healthy individuals, as evidenced by an AUC analysis (Fig. S4B ).

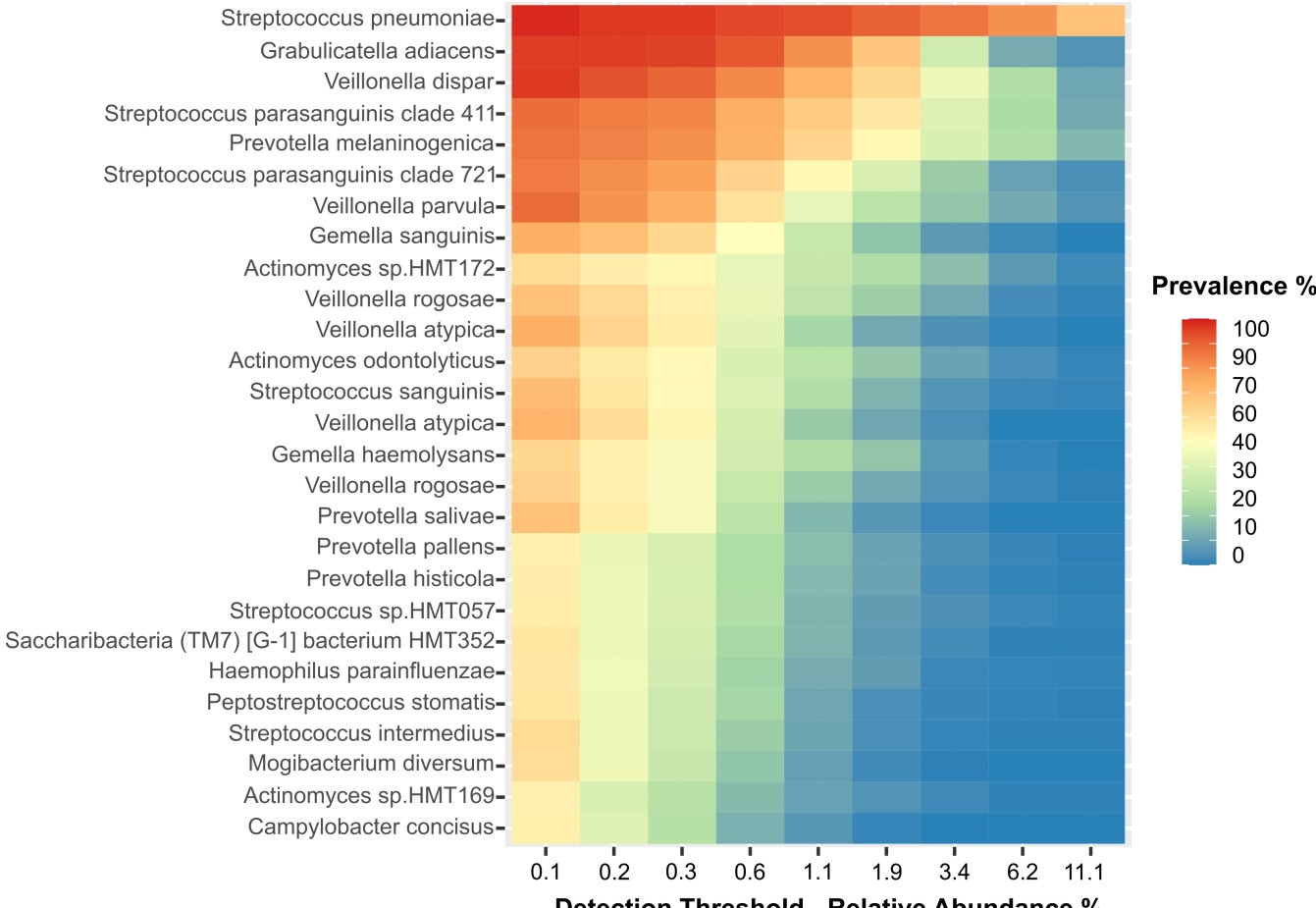

**FIG 4** Determination of the core microbiome for all clinical groups. Heatmap of relative abundance for each species is described on the y-axis. The x-axis shows prevalence of each relative abundance.

## Analysis of networking in bacterial communities shows marked contrasts in the different clinical groups

The interaction between the members of the bacterial community in each group was studied by network analyses. In each network, the size of the circle is proportional to the abundance of the species and each circle's color represents subclusters of bacteria with a stronger interaction between them. The color of each connection (edge) relates to the type of interaction, green is positive, and red is negative, results are presented in Fig. 5 and 6.

Of note, *Streptococcus pneumoniae* was by far the most abundant species in all five clinical groups, whereas abundance of other species varied per group (Fig. 5). In the AC group, all species showed about the same relative abundance, except *Streptococcus pneumoniae* that clearly had the highest abundance. When compared the AC group with AN several species of a green subcluster presented higher abundance including *Actinomyces odontolyticus*, *Streptococcus parasanguinis*, *Actinomyces graevenitzii*,

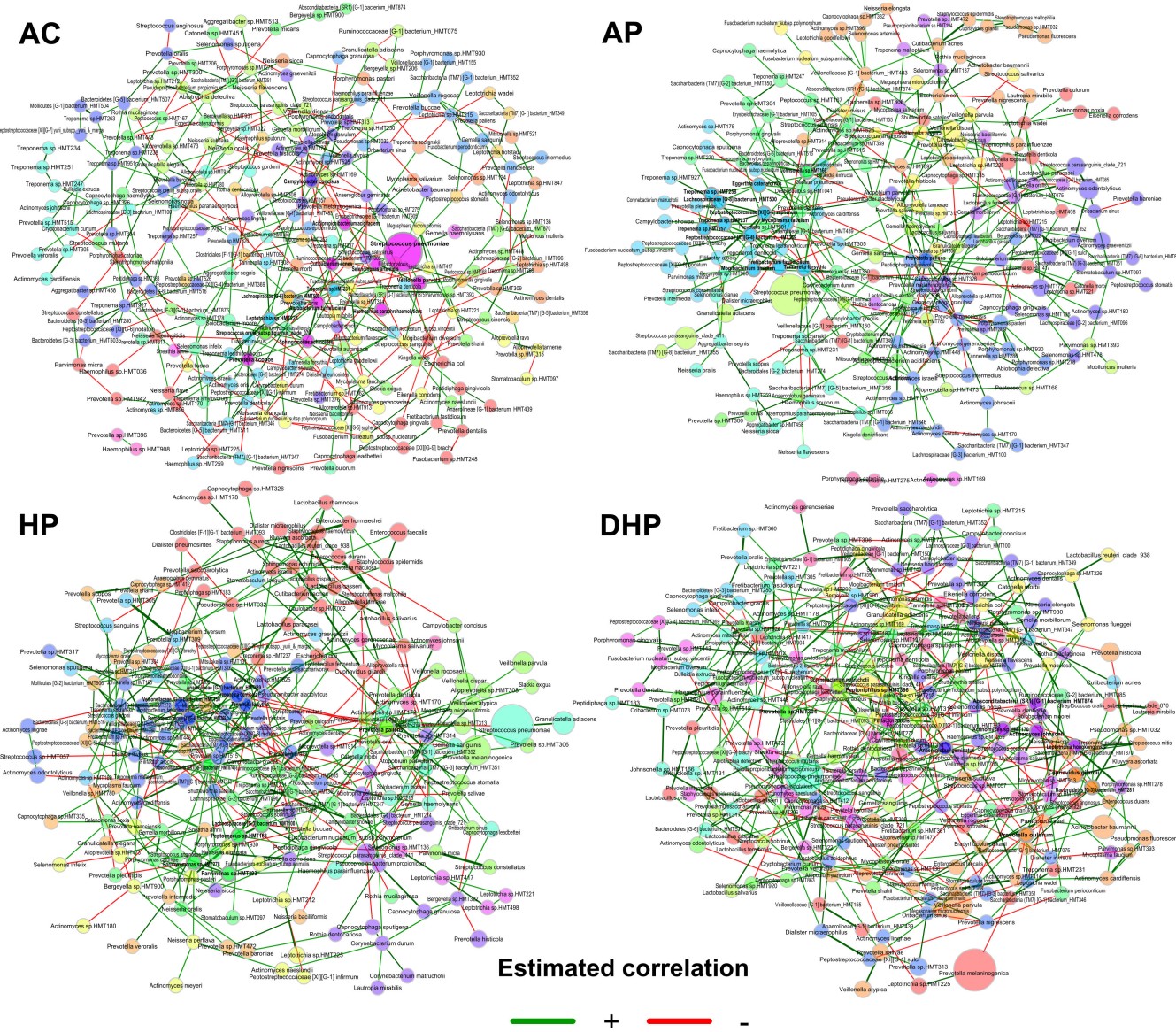

**FIG 5** Co-abundance networks for each clinical group analyzed with the SparCC method in the NetCoMi package. The size of the nodes was adjusted by a normalization of the counts, and a color was assigned to each subcluster within the network. Positive associations are shown with green connections and negative correlations with red connections.

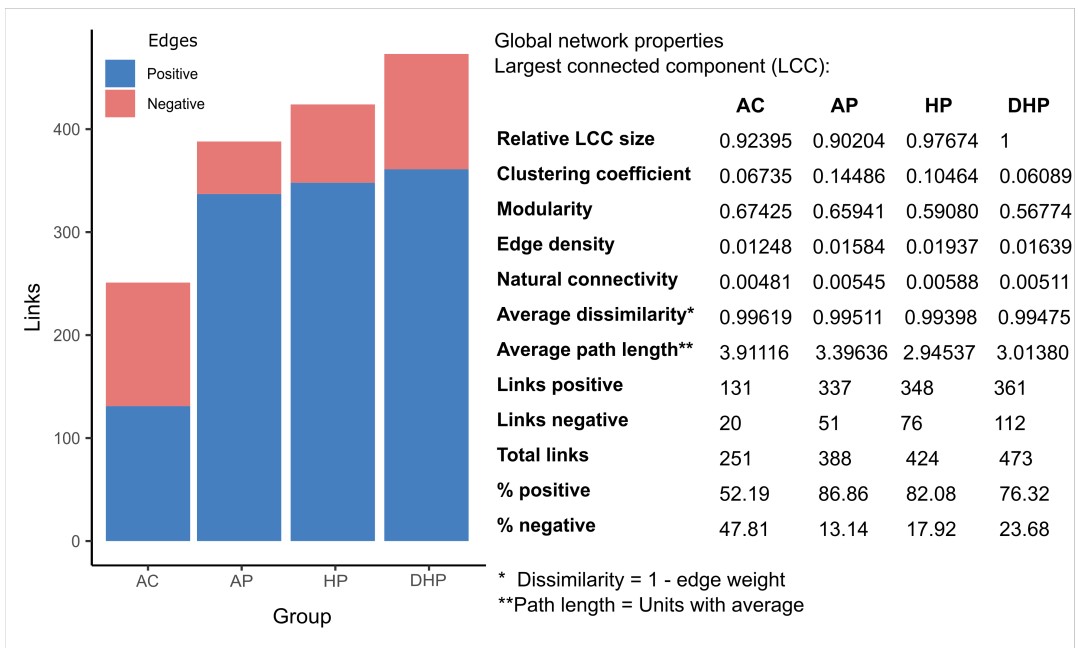

**FIG 6** Number and type of links (edges) in the bacterial network in saliva of patients from different clinical groups.

*Veillonela dispar,* and *Prevotella melaninogenica*. In the AP patients species of different subclusters were increased in abundance like *Veillonella dispar*, *Actinomyces odontolyticus*, *Prevotella melaninogenica,* and *Streptococcus parasanguinis*. In the HP, members of a green subcluster also increased, with a notorious abundance of *Prevotella melaninogenica*, followed by *Veillonella dispar*, *Veillonella parvula,* and *Granulicatella adiacens*. Finally, in the deceased patients (DHP) *Prevotella melaninogenica* becomes more abundant than *Streptococcus pneumoniae*, and *Granulicatella adiacens* also showed a marked abundance increase. Interestingly, *Acinetobacter baumannii* also presented increased abundance in this group and it showed positive association with *Kluyvera ascorbat*a and with *Ruminococcaceae* HMT075.

Members of the hub-species (species with more links and important to maintain the structure of the net) markedly varied between groups and only a few were present in more than one group, *Prevotella pallens* in the three sARS-CoV-2 + groups, AP, HP and DHP, *Tannerella forsythia* in the AP and HP groups, and *Absconditabacteria* HMT874 in the two severe groups HP and DHP (Fig. 5). The number of hub-species varied between 11 and 14 among the groups and were usually included within a single subcluster. Because of the marked abundance of *Streptococcus pneumoniae* in all groups we asked if it was linked to other species (Table S3). *Streptococcus pneumoniae* had positive links with *Granulicatella adiacens* in the AN, AP and HP patients but not in the DHP group, with *Gemella sanguinis* in the three sARS-CoV-2 + groups, AP, HP and DHP and with *Gemella haemolysans* in the two sARS-CoV-2- groups, AC and AN. *Streptococcus pneumoniae* also showed positive or negative links with different species in each clinical group (Table S3).

Finally, the number and type of total links also varied among groups (Fig. 6) and the total number of links gradually increased from healthy AC (251) to mild AP (388) to severe HP (424) and to deceased DHP (473). Whereas the type of association was rather balanced in the AC (52%+, 48%−) this balance was lost in all disease groups, where positive links increased to around 80%. Other properties of the nets are described in Fig. 6, connectivity values increase in moderate and severe cases, modularity decreased in severe cases, whereas clustering coefficient increase in moderate cases.

## Estimation of metabolic activity shows a highly increased activity by the bacterial community of the severe and deceased patients

An approximation to the metabolic activity of the bacterial community in each group was deduced with the use of Picrust2 software. Notably, the analyses revealed a significantly increased metabolic activity by the bacterial community of deceased patients (DHP) in most of the metabolic pathways, which gradually decreased in hospitalized patients, in ambulatory patients and finally in asymptomatic adults (Fig. 7). This drastic tendency is clearly illustrated looking at the metabolic pathways at the two different levels presented in Fig. 7A and B. The few activities diminished in severe (HP) and deceased patients (DHP) were the biosynthesis of the antimicrobial aminoglycosides (produced by *Actinomycetes*) and clavulanic acid (*Streptomyces*) as well as the metabolism of xenobiotics by P450 and polyketide sugar biosynthesis (Fig. 7B). Of note, microbiota of deceased patients presented a marked increased (z-score > 1.5) in degradation of organic compounds (caprolactam, fluorobenzoate, bisphenol, geraniol, naphthalene, and nitrotoluene) with environmental and health importance.

## DISCUSSION

The oral cavity may be the entry to the respiratory tract and the source of the oral microbiome of the upper and lower airways, including the lungs. In fact, the oral mucosa is recognized as an important site for SARS-CoV-2 infection and as a source for spreading the infection to the upper and lower respiratory tract (8). Thus, it becomes relevant to study the oral microbiome in patients with COVID-19 to try to elucidate its role in the severity of the disease. In this study we used saliva as a surrogate of the oral microbiota (14).

Our results show that diversity of the bacterial communities in saliva decreases as the severity of the disease increases, from ambulatory to hospitalized to deceased patients. Previous studies reported similar results when comparing healthy controls vs COVID-19 patients (10) or vs patients with long-COVID (9), but these studies did not contrast oral microbiota in patients with different severity of disease. Several studies in nasopharyngeal samples have consistently reported reduced diversity in COVID-19 patients (5), (6, 7) indicating infection is associated with important changes in the structure of bacterial populations colonizing the upper airways. To further study the nature of the changes in the microbiota we searched for differences in its composition among the groups. A main difference of our work with previous COVID reports is that we amplified the V1-V3 region of the 16S rRNA gene, which results in improved sensitivity when working with oral microbiome (38) and we used the updated eHOMD database with an extended coverage of oral species (20). eHOMD is considered as a comprehensive microbiome database with high resolution to study the human aerodigestive tract in health and disease usually to the level of species, that performs as well or better than other commonly used 16S databases (39). Thanks to this we were able to identify most ASV to the level of species, which contrasts with previous studies usually reporting down to the level of genus. This is relevant if we consider that the oral cavity hosts over 1,000 bacterial species (11) and reporting differences at the level of genus fell short in the interpretation of results. This can be illustrated by our findings that *Prevotella intermedia* and *Prevotella oris* were found significantly associated with asymptomatic adults, whereas *Prevotella melaninogenica* and *Prevotella salivae* were associated with deceased cases. These differences would be missed if analysis is limited to the genus level (see also Fig. S5).

Volcano test, random forest, and LEfSe analyses showed a strong agreement in severity-associated species and led us to identify specific changes in the composition of bacterial communities that differ in patients according to the severity of the disease. Thus, our results show that *Gemella haemolysans*, *Neisseria sicca*, *Prevotella oris*, *Prevotella nigrescens*, *Rothia dentocariosa*, *Porphyromonas gingivalis*, and *Granulicatella elegans* are salivary markers of asymptomatic adults. Previous studies have also reported *Granulicatella elegans* more prevalent in healthy subjects. *Porphyromonas gingivalis* is associated with periodontal disease and is considered a "keystone pathogen" because of its ability

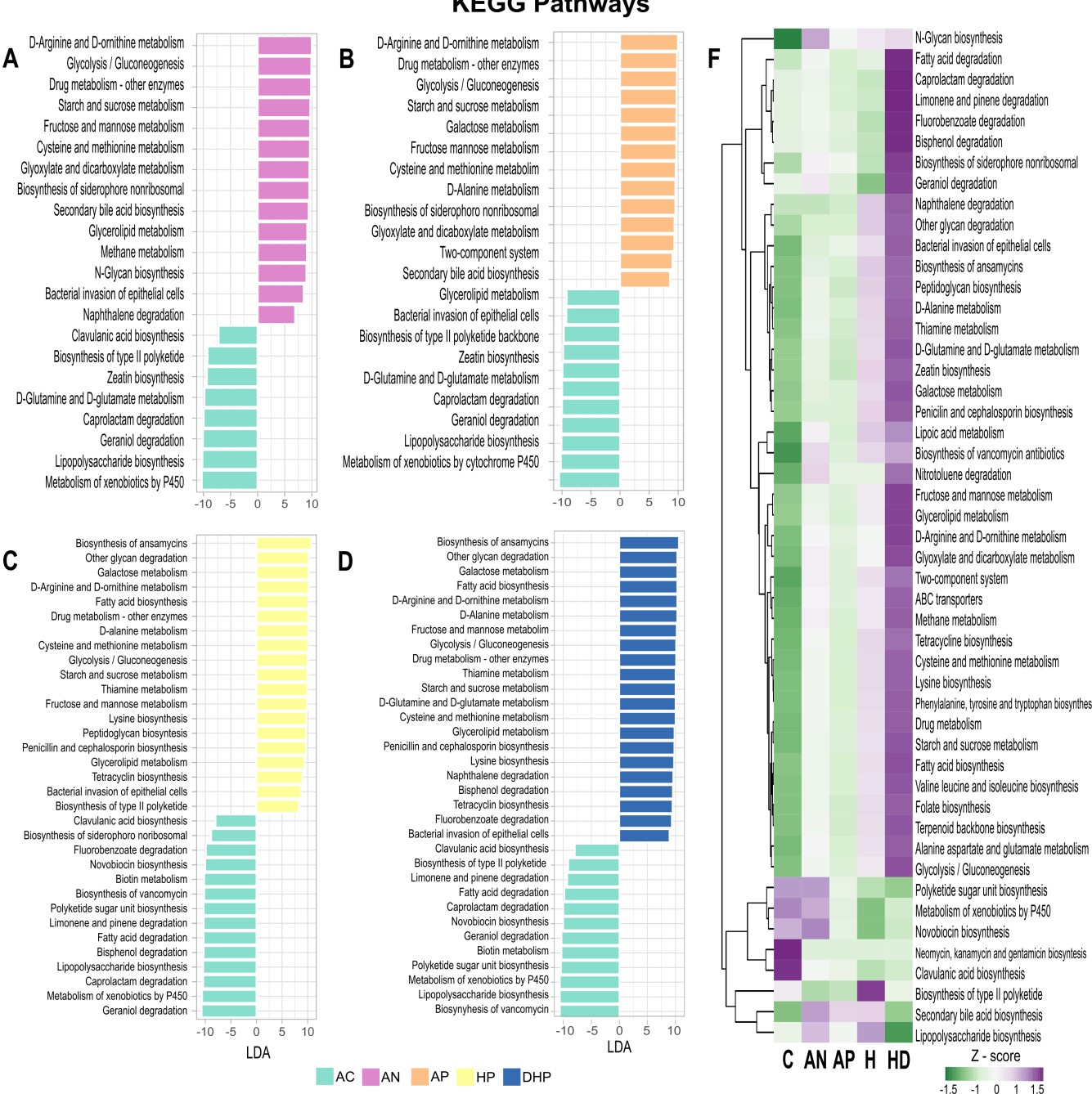

**FIG 7** Metabolic activity by the bacterial community present in each clinical group as inferred by Picrust2 software and KEGG database. (A), (B), (C), and (D) differentially abundant pathways between clinical groups. Pathways enriched in the AC group are indicated with negative LDA score (mint green) and pathways enriched in symptomatic groups with a positive score. Only pathways with a significant LDA threshold of > 5 are shown; (F) presents heatmap with z-scores of the relative abundance of the metabolic pathways level 3 as defined in KEGG.

to modify the host inflammatory response and cause dysbiosis. *Streptococcus parasanguinis*, *Actinomyces odontolyticus*, *Oribacterium sinus,* and *Atopobium parvulum* were characteristic of mild COVID cases, *Leptotrichia sp* HMT225, *Staphylococcus epidermidis,* and *Escherichia coli* were more abundant in severe cases and represent Taxa uncommon in healthy oral microbiota but rather opportunistic pathogens that overgrow when microbiota dysbiosis occur (40). Of note, in patients recovering from COVID-19, *Leptotrichia* has been reported to significantly decreased (10). We also found that *Prevotella*

*melaninogenica*, *Veillonella parvula*, *Granulicatela adiacens*, *Acinetobacter baumannii*, *Prevotella salivae*, *Leptotrichia wadei,* and *Capnocytophaga granulosa* were characteristic of fatal cases. *L. wadei* and *A. baumannii* are rare in healthy oral microbiota and are usually considered as co-infection, as described below. Of note, *Prevotella*, *Veillonella,* and *Leptotrichia* are high LPS-producers that promote inflammation and have been associated with long-lasting COVID (9) and interestingly, it has been postulated that SARS-CoV-2 may promote the growth of the anaerobic bacteria like *Prevotella, Veillonella,* and *Capnocytophaga* in the lungs and favor acute severe symptoms (41). The 2019 Wuhan outbreak could have been aggravated by *Prevotella*, which is aided by the coronavirus, possibly to adhere to epithelial cells (42).

A comparison with previous studies is difficult because most of them report up to genus level, but also because studies with oral microbiota are scarce (9) (10). Even studies with nasopharyngeal samples are limited and they show contradictory results. Whereas some agree reporting that *Corynebacterium* significantly decreased and *Prevotella* increased in infected patients (5) (6) (43), others have reported a reduction of *Prevotella* and *Veillonella* in COVID cases (7, 43). It is relevant to highlight the fact that in many instances disease associated bacteria are Taxa reported as members of the normal microbiota where abundance is modified and as reported here, networking is also altered. Changes in diversity lead to changes in networking patterns and this new scenario may favor persistence and abundance of certain Taxa that better adapt to the changing microbiota community. In this context, it was intriguing to see that microbiota of deceased patients presented a marked increased (Z-score > 1.5) in degradation of organic compounds (caprolactam, fluorobenzoate, bisphenol, geraniol, naphthalene and nitrotoluene) with environmental and health importance because they are man-made pollutants of high industrial production (44) (45) and probably bacteria are evolving to degrade these compounds (46).

Changes in the composition of bacterial communities are not the only factor to consider in microbiota studies. Coinfections with pathogens or opportunistic pathogens are common during viral pneumonia and known to increase the severity and risk of mortality (47). Thus, *Streptococcus pneumoniae* coinfection is a major cause of increase morbidity and mortality during influenza infection (48). Coinfections have also been documented in patients with COVID-19, particularly with *S. pneumoniae*, *K. pneumoniae* or *H. influenza* (49), and a metagenomic study of nasopharyngeal samples of patients with COVID-19 found a co-infection with a clinically relevant microorganism in 12.5% of patients (5). It should be noted that in our work *Streptococcus pneumoniae* was present in all groups studied and in fact its abundance was higher in healthy individuals and decreased as the severity of the disease increased, which questions its role as an opportunistic pathogen in our population.

In our study coinfections were common in severe hospitalized cases: *Escherichia coli*, *Leptotrichia* HMT225, and *Staphylococcus epidermidis* in severe patients and *Acinetobacter baumannii* and *Leptotricia wadei* in fatal cases. *Leptotrichia* has been found increased in patients with COVID-19 (10). *Leptotrichia* species are present in the oral cavity of healthy individuals and is considered an opportunistic pathogen because its abundance increases in caries, stomatitis or cases of septicemia in immunocompromised patients. *Leptotricia wadei* has been isolated in saliva of patients with caries or halitosis (40). On the other hand, *Acinetobacter baumannii*, a hospital-acquired opportunistic pathogen, has been reported in severe COVID patients, even in the lungs of fatal cases (50) and we found it significantly more abundant in the saliva of deceased patients.

We also determined the core microbiome for all clinical groups, which is seldom reported in microbiome studies. As indicated above, *Streptococcus pneumoniae* was found with high relative abundance in over 80% of all cases, representing the most prevalent and most abundant species in our population, regardless of the severity of COVID-19. The next most prevalent species included *Granulicatella adiacens*, *Veillonella dispar*, *Streptococcus parasanguinis,* and *Prevotella melaninogenica,* with a relative abundance of over 1.0% in most cases. Interestingly, *Granulicatella adiacens*, *Prevotella*

*melaninogenica,* and *Veillonella parvula* (all within the 10 most abundant in the core microbiome) significantly differentiated the deceased patients. At this point it is not possible to conclude whether they play a role in the progression to fatal cases or are present in all groups because they are resilient to microenvironment changes.

Microbiota form complex ecosystems on human surfaces reflecting strong positive or negative interactions and studies on these communities should not be limited to report differences in presence or abundance. Networks of co-abundance or co-dependency are necessary to better understand the role of microbiota in health and disease (51). Accordingly, we analyzed the interaction between members of the bacterial community in each clinical group and found marked differences. Each network was composed of subclusters where its members had strong positive interaction between them. The composition of these subclusters varied in each clinical condition. In asymptomatic individuals the microbial community showed a balanced interaction, 52%-positive, 48%-negative. In contrast, this balanced was lost in all disease groups and positive correlation was the more prevalent (around 80%). Although some species remain in all groups (as observed also with the core analyses), changes in the composition, abundance and even in the hub-species were apparent in each group. Some species showed a clear increase in relative abundance as the severity of disease progress, like *Prevotella melaninogenica*, *Veillonella dispar,* and *Granulicatella adiacens* and these changes were also associated with differences in hub-species.

The marked changes in the bacterial population of severe cases alter the integrity of the community. Furthermore, it is intriguing to observe that the number of links between member of the net increase as severity increases. Probably when homeostasis in the microbial community is broken, bacteria are looking for new partners and what we see is this acute search. Thus, major changes occurred in the structure of bacterial communities as the severity of the disease increased. The possible role of these large changes in the pathogenesis and severity of COVID-19 remains to be studied. Perhaps these changes affect the nature of the local and systemic inflammatory response.

Thus, the complex community of healthy people, balanced in its interactions, could be considered a normal state or normonetting. When the balance is gradually lost as severity of the disease increased the altered networking or disnetting may result in a highly unregulated community. A previous study also found that the complexity of co-abundance networks was decreased in patients with severe COVID-19, indicating a reduction in the interaction between members of the bacterial community (6). A highly unregulated bacterial community may result in marked metabolic alterations and strong microenvironment changes. In fact, our metabolic deductions showed significantly large changes in metabolic pathways as the severity of disease increased (Fig. 7). It was noticeable that the activity of most metabolic pathways was markedly and gradually increased from asymptomatic to ambulatory, to hospitalized and to deceased patients. This may suggest that the gradual looseness in negative interactions impacts metabolic activity of the bacterial community. The increased metabolites produced by the microbiota may have profound effects on the patient's health (12).

COVID-19 is a complex multisystemic, multiorgan disease probably due to the ability of the virus to disseminate and invade several cell types of the body (1, 2). Changes in the bacterial communities may also contribute to severity of the disease. Oral microbes and microbial molecules might directly enter the bloodstream and contribute to the pathogenesis of systemic diseases. Brain specimens and cerebrospinal fluid from individuals diagnosed with Alzheimer's disease suggest that *P. gingivalis* could colonize the brain and induce neurodegeneration (52). Also, network analyses in oral microbiota have shown a significant correlation of disease-associated bacterial species with proinflammatory cytokines (53), and a significant correlation of the abundance of *Staphylococcus* in the nasopharynx with systemic levels of IL-6 and TNF has been reported (43). On the other hand, oral microbiota plays integral roles in maintaining host health systemically and locally. One example is the conversion by oral microbes of nitrate to nitrite, which is absorbed and converted to NO, important for the control of blood

pressure and endothelial function (54). Thus, disruption of the oral microbiota may in several ways affect severity of COVID-19 disease.

Although there were important consistencies in results across methods, we still observed variation in taxa associated with disease severity in the different analytical methods. Currently, there is no consensus on the best methods to analyze microbiota and new approaches continue to be proposed, we choose to use several approaches including those most commonly reported. Sample size was also a limitation that affected the strength of the analyses, particularly for the group of deceased patients, and which in part was due to the lack of previous reports on the subject but also to the limitations in recruitment and sampling of patients during the first wave of the epidemy. Another limitation is that we used the updated eHOMD database for oral microbiome and our analysis was limited to the extent of this database; however, eHOMD is recognized as the most comprehensive reference for microbiome studies in the aerodigestive tract and commonly used for studies in the oral cavity. Furthermore, using this database we detected several opportunistic pathogens usually absent in the mouth like *Acinetobacter baumannii* or enterobacteria.

In summary, we report significant changes in diversity, composition, and networking in the saliva microbiota of patients with COVID-19 and found patterns associated with severity of the disease. We report oral species associated with each clinical stage because of its presence or abundance, as well as infection with opportunistic pathogens. Patterns of networking were also found associated with severity of disease. A highly regulated community (normonetting) was found in healthy people, whereas poorly regulated populations (disnetting) were characteristic of severe cases. Characterization of microbiota in saliva may offer important clues in the pathogenesis of COVID-19 and may also identify potential markers for prognosis of the disease.

## AUTHOR AFFILIATIONS

[1]Departamento de Bioquímica, Escuela Nacional de Ciencias Biológicas, Instituto Politécnico Nacional, México, Mexico

[2]Universidad Autónoma de Baja California Sur, La Paz, Baja California Sur, Mexico

[3]Centro de Investigaciones Biológicas del Noroeste SC, La Paz, Baja California Sur, Mexico

[4]Unidad de Investigación Médica en Enfermedades Infecciosas, UMAE Pediatría, Centro Médico Nacional SXXI, IMSS, Torreón, Mexico

[5]Laboratorio de Secuenciación, División de Desarrollo de la Investigación, IMSS, Torreón, Mexico

[6]División de Desarrollo de la Investigación en Salud, Coordinación de Investigación en Salud, IMSS, Torreón, Mexico

[7]División de Laboratorios Especializados, Coordinación de Calidad de Insumos y Laboratorios Especializados, IMSS, Torreón, Mexico

[8]Coordinación de Vigilancia Epidemiológica, Dirección de Prestaciones Médicas, IMSS, Torreón, Mexico

[9]Hospital General de Zona No. 27, Ciudad de México Norte, IMSS, México, Mexico

[10]Coordinadora de Información y Análisis Estratégicos, OOAD Cd de México, México, Mexico

[11]Hospital General de Zona con Medicina Familiar No. 8 Cd de México, IMSS, México, Mexico

[12]Departamento de Vigilancia Epidemiológica, IMSS Bienestar, México, Mexico

## AUTHOR ORCIDs

Javier Torres   http://orcid.org/0000-0003-3945-4221

## FUNDING

| Funder | Grant(s) | Author(s) |
|---|---|---|
| Consejo Nacional de Ciencia y Tecnología (CONACYT) | 312992 | Javier Torres |

## AUTHOR CONTRIBUTIONS

Violeta Larios Serrato, Data curation, Formal analysis, Methodology, Software | Beatriz Meza, Data curation, Formal analysis, Methodology | Carolina Gonzalez-Torres, Methodology, Validation | Javier Gaytan-Cervantes, Methodology, Validation | Joaquín González Ibarra, Methodology, Resources, Supervision | Clara Esperanza Santacruz Tinoco, Methodology, Resources, Supervision | Yu-Mei Anguiano Hernández, Investigation, Methodology, Resources | Bernardo Martínez Miguel, Methodology, Resources | Allison Cázarez Cortazar, Methodology, Resources | Brenda Sarquiz Martínez, Data curation, Methodology, Resources | Julio Elias Alvarado Yaah, Investigation, Methodology, Validation | Antonina Reyna Mendoza Pérez, Investigation, Methodology | Juan José Palma Herrera, Methodology, Resources, Validation | Leticia Margarita García Soto, Data curation, Methodology, Validation | Adriana Inés Chávez Rojas, Methodology, Resources | Guillermo Bravo Mateos, Formal analysis, Investigation, Methodology | Gabriel Samano Marquez, Data curation, Supervision, Validation | Concepción Grajales Muñiz, Conceptualization, Formal analysis, Funding acquisition, Supervision, Writing – original draft, Writing – review and editing | Javier Torres, Conceptualization, Formal analysis, Funding acquisition, Supervision, Writing – original draft, Writing – review and editing

## DATA AVAILABILITY STATEMENT

Database of microbiome (Taxa table) is registered at the NCBI with the BioProject ID PRJNA896341, in the link http://www.ncbi.nlm.nih.gov/bioproject/896341 and metadata of patients in the link https://github.com/BiosLS/covidimss.

## ADDITIONAL FILES

The following material is available online.

### Supplemental Material

**Fig. S1 (msystems.01062-22-s0001.tif).** Volcano analysis of microbiota in patients with mild disease.
**Fig. S2 (msystems.01062-22-s0002.tif).** ROC curve showing significance in microbial composition between group.
**Fig. S3 (msystems.01062-22-s0003.tif).** Relative abundance of species significantly different between groups.
**Fig. S4 (msystems.01062-22-s0004.tif).** Score of dysbiosis in symptomatic patients SARS-CoV-2 infected.
**Fig. S5 (msystems.01062-22-s0005.tif).** Stack bar of taxa in each clinical group at the level of family and species.
**Table S1 (msystems.01062-22-s0006.docx).** List of amplification primers.
**Table S2 (msystems.01062-22-s0007.docx).** List of adapter primers.
**Table S3 (msystems.01062-22-s0008.docx).** Species linked to S. pneumoniae.
**Legends (msystems.01062-22-s0009.docx).** Legends to supplemental figures and table.

### Open Peer Review

**PEER REVIEW HISTORY (review-history.pdf).** An accounting of the reviewer comments and feedback.

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
