## [Reviewer comments · mSystems]

Diversity, composition, and networking of saliva microbiota distinguish the severity of COVID-19 episodes as revealed by an analysis of 16S rRNA variable V1-V3 regions sequences

Javier Torres, Violeta Larios, Beatriz Meza, Carolina Gonzalez-Torres, Javier Gaytan-Cervantes, Joaquin González-Ibarra, Clara Santacruz-Tinoco, Yu-Mei Anguiano-Hernández, Bernardo Martinez-Miguel, Allison Cazarez-Cortazar, Brenda Sarquiz-Martinez, Julio Alvarado-Yaah, Antonina Mendoza-Perez, Juan Palma-Herrera, Leticia Garcia-Soto, Adriana Chavez-Rojas, Guillermo Bravo-Mateos, Gabriel Samano-Marquez, and Concepción Grajales-Muñiz

Corresponding Author(s): Javier Torres, Mexican Institute of Social Security

Review Timeline:

Submission Date:	November 2, 2022
Editorial Decision:	January 5, 2023
Revision Received:	April 10, 2023
Accepted:	April 17, 2023

Editor: Marc Cook

Reviewer(s): Disclosure of reviewer identity is with reference to reviewer comments included in decision letter(s). The following individuals involved in review of your submission have agreed to reveal their identity: Jarrad Hampton-Marcell (Reviewer #1); Valerio Iebba (Reviewer #2)

Transaction Report:

DOI: <https://doi.org/10.1128/msystems.01062-22>

January 5, 2023

Prof. Javier Torres
Mexican Institute of Social Security (IMSS)
Infectious Diseases
Av. Cuauthemoc 330
Av. Cuitlahuac # 2931 int 16
Mexico, D.F 92840
Mexico

Re: mSystems01062-22 (Diversity, composition, and networking of saliva microbiota distinguish the severity of COVID-19 episodes as revealed by an analysis of 16S rRNA variable V1-V3 regions sequences)

Dear Prof. Javier Torres:

Thank you for submitting your manuscript to mSystems. We have completed our review and I am pleased to inform you that, in principle, we expect to accept it for publication in mSystems. However, acceptance will not be final until you have adequately addressed the reviewer's comments.

Preparing Revision Guidelines

Sincerely,

Marc Cook

Editor, mSystems

Journals Department

Reviewer comments:

Reviewer #1 (Comments for the Author):

The author details the connection between the oral microbiota across severity in COVID-19 infections. This manuscript is timely and needed given the need to better understand impact of hospitalizations and co-infections during the pandemic, as well as health outcomes. I offer the following comments below:

Line 136: It is not clear whether the authors received IRB approval to conduct the study.

Line 206: It is not clear why the author chose to use NMDS versus principal coordinate analysis (PCoA) when assessing and visualizing microbial community structure. This should be further explained in the methods section. Furthermore, the stress value is often reported when using NMDS as this signifies whether the ordination is arbitrary i.e., a stress value above .02 is generally questionable typically.

Line 213: For the random forest model, the author should clarify whether an out-of-bag or a cross-folded validation was applied. These parameters can drastically change the model's accuracy and reliability. Additionally, was the random forest model generated in R or python?

Line 266: The author mentions some separation in of bacterial communities, but this is not clear. Is the author referring to microbial variation? If so, what is the R-value associated with this variation and is this significant. This should be added to provide clarity. Additionally, the author previously mentioned in the discussion section applying permutational dispersion; however, this is not reported in the results. Is there homogeneity among the samples?

Lines 271-279: The author list numerous species that appear to alter among groups. Yet, there is no mention of whether abundances are significant different among groups. Furthermore, it is not clear what test is being used to identify targeted microbial taxa. Are species lowly abundant? Are rank correlations being implemented or ANCOM. Also, it would be beneficial to provide a general distribution of microbial phyla. It is not clear whether the species reside in Firmicutes, Actinobacteria, or Bacteroidetes etc.

Line 294: It would be helpful to report effect size for microbial taxa significantly different between groups.

Line 364: It is standard to report the area under curve (AUC) for random forest in order to better assess the model's accuracy. In addition, it would be beneficial to report the confusion matrix in order to better assess how many samples were correctly predicted for each group.

Line 388: What is meant by endurance?

Lines 389-392: It is not exactly clear what the implications of the microbial taxa regarding the core microbiome. Does the core microbiome increase/decrease with severity. Is this related to the previously mentioned alpha diversity measures in any way? This is not entirely clear.

Line 415: The author denotes significance, but it is not evident in Table 2. It would be helpful to show p-values for Pearson correlations. Also, I think the x and y-axis for Table 2 are inverted. As currently constructed it comes off as comparing positive and negative links, but I believe you are truly comparing links between groups which would then provide the opportunity to add an additional column to report p-values.

Line 425: It would be helpful to report centrality to support this statement.

Line 430: How were subclusters identified? This wasn't stated in methods. Did you use k-means clustering when generating networks or are you basing this on a network property such as closeness. This should be explained and reported either way.

Lines 439-442: It would be helpful to define how you are grouping the microbial taxa into clusters.

Line 501: What test was applied to denote significance?

Line 514: For Figure 6A, what does the x-axis represent?

Lines 584-589: I agree with the author that comparison of microbial taxa across severity is rare and makes it difficult to compare to previous studies, but it would be helpful to offer an explanation as to why certain bacteria are favored for a given state. The author does this for the asymptomatic state, it would be helpful to add some clarity as the results section list a number of different species.

Lines 643-654: I'm not exactly how this is accurate. The author did not assess connectivity i.e., if a portion of the network is disrupted how many links are loss within the network.

Additional comments:

It would be helpful to discuss the limitations of the study. Specifically, the number of statistically significant bacteria across severity. While there are some common bacteria listed across statistical analyses, there are some bacteria that are significant in some test but not others. This should be addressed, which is likely due to variation among humans.

Given the author repeated mentioned pathogenesis and dysbiosis in the manuscript, I assumed there would be some type of dysbiosis analysis. This would complement the analyses and help discern whether the oral microbiota is associated with severity. DysbiosisR seems like an appropriate statistical analysis given the methodology of the paper.

Reviewer #2 (Comments for the Author):

-line 50: please add number of patients (n=xxx) for each group.

-line 138: provide PDF copy of Ethical Committee approval.

-line 200: please report how relative abundances matrix was normalized before stepping to alpha- and beta-diversity calculations.

-line 211: without a proper normalization (CLR, Gaussain-like/standardization) differences in taxa are flawed by inherent compositionality.

-line 219: this paragraph is not clear. Definite prevalence and abundance cutoffs were chosen prior to analyze alfa- beta- diversity and the others analysis? Please explain.

-line 226: replace "co-occurrence" with "co-abundance" throughout the manuscript.

-line 235: as for microbiota data, also Picrust2 data need to be transformed before making any statistical operation, please write how data were transformed (see advise by Douglas and Langille).

-line 241: Bioproject not visible, please ensure availbaility.

-line 266: please add 95% ellipse CI for each group.

-line 266: please report for all the beta-diversity metrics the P values for inter-group comparison.

-line 268: this graph is not meaningful to the readers, because no insights are given of DA among discriminant species (severity-driven P values ?). Please do a RF or VIP plot with variable importance. Or, at least draw P values for each species among all groups (Kruskal-Wallis) with post-hoc test (Mann-Whitney), but on transformed data!

-line 270: I advise to have an insight on 16S NCBI database, encompassing >50k genomes, updated every two-three days. In contrast, in eHOMD v3 (is that this version was used?) total number of genomes are 2087 including non-oral/non-nasal taxa. It means that some non-oral bacterial species should have been unrecognized or under-detected.

-line 279: please report objective values of Delta among relative abundances or P values when dealing with comparisons, avoid subjective jargon such as "particularly higher" and so on.

-line 296: why enhanced? In any case, Volcano plots do not visualize species with zero relative abundance in at least one group. I advise to draw a Figure1D panel reporting FoldC differences. Or, at least, a NMDS bi-plot, superimposing on Figure1B the eigenvalues of bacterial drivers, we should see on bottom-right corner species such as *A.baumannii*, *Prevotella* spp., *Veillonella* spp., etc..

-line 363: please cluster the bacterial species by the cohorts, reporting in descending order of "mean accuracy" and "LDA": in this manner it's easier for the Reader to follow results in a nutshell. I would also like to see a ROC curve with selected bacterial species able to separate 1vs1 the different cohorts.

-line 413-414: this is not the usual workflow for network co-abundance analysis. The best option is to do networks with all the species, then filtering edges with Pearson coefficients (e.g. 0.7) or P values (e.g. <0.01), then ruling out singletons, then assessing communities/clusters. In this way the network is unbiased for community and functional definition. I advise to calculate betweenness centrality and figure out keystone species in each cohort. I stress also the previous comment on drawing ROC curves using the selected bacterial species (here is right to use results from volcano, lfeSe, RF, and networks).

-line 419: transform this Table2 in an inset of Figure5 as stacked barplot. Erase then Table2.

-line 501: the Picrust2 dataset was normalized for total gene count before doing analyses? it seems to be a bias at level2, which affects also level3.

-line 561: see previous comment about eHOMD.

-line 645: I would see also these network metrics computed for each severity-cohort and reported in Figure5: i) Average Path length; ii) density; iii) diameter; iv) number of communities.

The author details the connection between the oral microbiota across severity in COVID-19 infections. This manuscript is timely and needed given the need to better understand impact of hospitalizations and co-infections during the pandemic, as well as health outcomes. I offer the following comments below:

Line 136: It is not clear whether the authors received IRB approval to conduct the study.

Line 206: It is not clear why the author chose to use NMDS versus principal coordinate analysis (PCoA) when assessing and visualizing microbial community structure. This should be further explained in the methods section. Furthermore, the stress value is often reported when using NMDS as this signifies whether the ordination is arbitrary i.e., a stress value above .02 is generally questionable typically.

Line 213: For the random forest model, the author should clarify whether an out-of-bag or a cross-folded validation was applied. These parameters can drastically change the model's accuracy and reliability. Additionally, was the random forest model generated in R or python?

Line 266: The author mentions some separation in of bacterial communities, but this is not clear. Is the author referring to microbial variation? If so, what is the R-value associated with this variation and is this significant. This should be added to provide clarity. Additionally, the author previously mentioned in the discussion section applying permutational dispersion; however, this is not reported in the results. Is there homogeneity among the samples?

Lines 271-279: The author list numerous species that appear to alter among groups. Yet, there is no mention of whether abundances are significant different among groups. Furthermore, it is not clear what test is being used to identify targeted microbial taxa. Are species lowly abundant? Are rank correlations being implemented or ANCOM. Also, it would be beneficial to provide a general distribution of microbial phyla. It is not clear whether the species reside in Firmicutes, Actinobacteria, or Bacteroidetes etc.

Line 294: It would be helpful to report effect size for microbial taxa significantly different between groups.

Line 364: It is standard to report the area under curve (AUC) for random forest in order to better assess the model's accuracy. In addition, it would be beneficial to report the confusion matrix in order to better assess how many samples were correctly predicted for each group.

Line 388: What is meant by endurance?

Lines 389-392: It is not exactly clear what the implications of the microbial taxa regarding the core microbiome. Does the core microbiome increase/decrease with severity. Is this related to the previously mentioned alpha diversity measures in any way? This is not entirely clear.

Line 415: The author denotes significance, but it is not evident in Table 2. It would be helpful to show p-values for Pearson correlations. Also, I think the x and y-axis for Table 2 are inverted. As currently constructed it comes off as comparing positive and negative links, but I believe you are truly comparing links between groups which would then provide the opportunity to add an additional column to report p-values.

Line 425: It would be helpful to report centrality to support this statement.

Line 430: How were subclusters identified? This wasn't stated in methods. Did you use k-means clustering when generating networks or are you basing this on a network property such as closeness. This should be explained and reported either way.

Lines 439-442: It would be helpful to define how you are grouping the microbial taxa into clusters.

Line 501: What test was applied to denote significance?

Line 514: For Figure 6A, what does the x-axis represent?

Lines 584-589: I agree with the author that comparison of microbial taxa across severity is rare and makes it difficult to compare to previous studies, but it would be helpful to offer an explanation as to why certain bacteria are favored for a given state. The author does this for the asymptomatic state, it would be helpful to add some clarity as the results section list a number of different species.

Lines 643-654: I'm not exactly how this is accurate. The author did not assess connectivity i.e., if a portion of the network is disrupted how many links are lost within the network.

Additional comments:

It would be helpful to discuss the limitations of the study. Specifically, the number of statistically significant bacteria across severity. While there are some common bacteria listed across statistical analyses, there are some bacteria that are significant in some test but not others. This should be addressed, which is likely due to variation among humans.

Given the author repeatedly mentioned pathogenesis and dysbiosis in the manuscript, I assumed there would be some type of dysbiosis analysis. This would complement the analyses and help discern whether the oral microbiota is associated with severity. DysbiosisR seems like an appropriate statistical analysis given the methodology of the paper.

Reviewer #1

The author details the connection between the oral microbiota across severity in COVID-19 infections. This manuscript is timely and needed given the need to better understand impact of hospitalizations and co-infections during the pandemic, as well as health outcomes. I offer the following comments below:

Line 136: It is not clear whether the authors received IRB approval to conduct the study.
R. The study was approved by the Institutional IRB, this is now described in lines 145-147

Line 206: It is not clear why the author chose to use NMDS versus principal coordinate analysis (PCoA) when assessing and visualizing microbial community structure. This should be further explained in the methods section. Furthermore, the stress value is often reported when using NMDS as this signifies whether the ordination is arbitrary i.e., a stress value above .02 is generally questionable typically.

R. NMDS with Bray-Curtis dissimilarity index is suggested to visualize beta diversity, although the use of PCoA is usually more accepted. Thus, as suggested, we now show the analyses of beta diversity using principal coordinate analysis (PCoA) with unweighted and weighted UniFrac to visualize covid-associated bacterial communities. See modified figure 1B.

Line 213: For the random forest model, the author should clarify whether an out-of-bag or a cross-folded validation was applied. These parameters can drastically change the model's accuracy and reliability. Additionally, was the random forest model generated in R or python?

R. For the Random Forest, we used the R Caret's package, MLeval, running 1000 trees. For cross-validation values were multiplied by 10, and the training set was 90% while the test set was 10%. In addition, for further validation an AUC frame was determined (see lined 231-236 and suppl figure 2).

Line 266: The author mentions some separation in of bacterial communities, but this is not clear. Is the author referring to microbial variation? If so, what is the R-value associated with this variation and is this significant. This should be added to provide clarity. Additionally, the author previously mentioned in the discussion section applying permutational dispersion; however, this is not reported in the results. Is there homogeneity among the samples?

R. The significance in the separation of bacterial communities was analyzed by PERMANOVA, ANOSIM and PERMDISP tests as described in Methods, lines 219-22 and Results, in figure 1B.

Lines 271-279: The author list numerous species that appear to alter among groups. Yet, there is no mention of whether abundances are significant different among groups. Furthermore, it is not clear what test is being used to identify targeted microbial taxa. Are species lowly abundant? Are rank correlations being implemented or ANCOM. Also, it

would be beneficial to provide a general distribution of microbial phyla. It is not clear whether the species reside in Firmicutes, Actinobacteria, or Bacteroidetes etc.

R. Considering that variation of species among groups is already analyzed more in detail by enhanced volcano, Random Forest and Lefseq, we decided to remove the initial analyses previously shown in figure 1C.

Line 294: It would be helpful to report effect size for microbial taxa significantly different between groups.

R. To analyze the differences in taxa abundance among groups, the Fold Change (FC) and the FDR-adjusted values were calculated using DESeq2, and for normalization geometric mean was calculated for each ASVs across all samples. The data were previously filtered based on the significant results of RandomForest and Lefseq. This is now described in Methods, lines 237-242.

Line 364: It is standard to report the area under curve (AUC) for random forest in order to better assess the model's accuracy. In addition, it would be beneficial to report the confusion matrix in order to better assess how many samples were correctly predicted for each group.

R. The AUC and the confusion matrix for the random forest is now shown in Suppl. figure 2.

Line 388: What is meant by endurance?

R. By endurance we mean taxa that persist the most across all clinical groups, regardless of the changes in the microenvironment in each clinical group. See note in line...

Lines 389-392: It is not exactly clear what the implications of the microbial taxa regarding the core microbiome. Does the core microbiome increase/decrease with severity. Is this related to the previously mentioned alpha diversity measures in any way? This is not entirely clear.

R. The core genome estimation is meant to find out which taxa prevail the most in samples from all groups analyzed after filtering data. As mentioned in Methods lines 246-249 we filtered using taxa prevalent in over 50% of the samples with a relative abundance above 0.2. These are the Taxa that vary the least across clinical groups and are not meant to give data on increase/decrease with severity nor with diversity.

Line 415: The author denotes significance, but it is not evident in Table 2. It would be helpful to show p-values for Pearson correlations. Also, I think the x and y-axis for Table 2 are inverted. As currently constructed it comes off as comparing positive and negative links, but I believe you are truly comparing links between groups which would then provide the opportunity to add an additional column to report p-values.

R. Table 2 was questioned by the two reviewers and one suggested presenting the data as a figure, and so, figure 6 now replaces Table 2. The significance data are described in lines 476-482 and in figure 6.

Line 425: It would be helpful to report centrality to support this statement.

R. Additional values to describe the nets are now included in figure 6.

Line 430: How were subclusters identified? This wasn't stated in methods. Did you use k-means clustering when generating networks or are you basing this on a network property such as closeness. This should be explained and reported either way.

R. Clusters were identified by NetCoMi using greedy modularity optimization, by cluster_fast_greedy and illustrated using the igraph R package <https://github.com/stefpeschel/NetCoMi>. The method is described in more detail in lines 258-262.

Lines 439-442: It would be helpful to define how you are grouping the microbial taxa into clusters.

R. see answer to line 430 for details.

Line 501: What test was applied to denote significance?

R. As described in the above comment, in the co-occurrence network analysis NetCoMi defines subclusters based on Pearson correlation to show species with significant correlation as described by color of the nodes. This is described in Methods, lines 251-256.

Line 514: For Figure 6A, what does the x-axis represent?

R. The x-axis in Figure 7A-D (previously 6A) represents pathway abundance as the legend describes now. Please note that the figure was edited according to reviewer's suggestions

Lines 584-589: I agree with the author that comparison of microbial taxa across severity is rare and makes it difficult to compare to previous studies, but it would be helpful to offer an explanation as to why certain bacteria are favored for a given state. The author does this for the asymptomatic state, it would be helpful to add some clarity as the results section list a number of different species.

R. This is a good point that we tried to address in changes made in lines 650-663 of the Discussion section.

Lines 643-654: I'm not exactly how this is accurate. The author did not assess connectivity i.e., if a portion of the network is disrupted how many links are loss within the network.

R. What we want to stress here is the changes in the proportion of positive and negative interactions observed between microbiota members of the control group and the group of deceased patients. We do have the data on the variation of number of links among the different groups. Additional properties of the global network are presented in Figure 6 (that related Table 2) to more clearly show the contrast between groups including the total number of links.

Additional comments:

It would be helpful to discuss the limitations of the study. Specifically, the number of statistically significant bacteria across severity. While there are some common bacteria

listed across statistical analyses, there are some bacteria that are significant in some test but not others. This should be addressed, which is likely due to variation among humans.

R. A paragraph with limitations of the study was added in the Discussion section, lines 680-692.

Given the author repeated mentioned pathogenesis and dysbiosis in the manuscript, I assumed there would be some type of dysbiosis analysis. This would complement the analyses and help discern whether the oral microbiota is associated with severity. Dysbiosis.

R. A dysbiosis score was calculated using dysbiosisR V1.0.4 and results are presented in Suppl fig 4. The analysis is also described in Methods lines 224-227 and in Results lines 311-314.

Reviewer #2 (Comments for the Author):

-line 50: please add number of patients (n=xxx) for each group.

R. numbers were added as suggested.

-line 138: provide PDF copy of Ethical Committee approval.

R. PDF will be uploaded as requested.

-line 200: please report how relative abundances matrix was normalized before stepping to alpha- and beta-diversity calculations

R. For normalization geometric mean was calculated for each ASVs across all samples using Total Sum Scaling (TSS) as described in Methods lines 209-211, where a reference was also added, ref 21. TSS is recommended because it most accurately captures the composition of the original communities, whereas UQ and CSS distort communities.

-line 211: without a proper normalization (CLR, Gaussain-like/standardization) differences in taxa are flawed by inherent compositionality.

R. see comment above.

-line 219: this paragraph is not clear. Definite prevalence and abundance cutoffs were chosen prior to analyze alfa- beta-diversity and the others analysis? Please explain.

R. The paragraph on the analyses of diversity, lines 209-222 was edited to address this and other observations.

-line 226: replace "co-occurrence" with "co-abundance" throughout the manuscript.

R. done, thanks

-line 235: as for microbiota data, also Picrust2 data need to be transformed before making any statistical operation, please write how data were transformed (see advise by Douglas and Langille).

R. We also edited the paragraph on analyses of metabolic routes, lines 267-277 to better describe the methods.

-line 241: Bioproject not visible, please ensure availability.

R. Data will be released on the 2023-11-20.

-line 266: please add 95% ellipse CI for each group.

R. By suggestion of reviewer 1 this analysis was modified as illustrated in Figure 1B where some statistical values are shown. This is also described in Methods, lines 216-222.

-line 266: please report for all the beta-diversity metrics the P values for inter-group comparison.

R. p-values are now shown in Figure 1B

-line 268: this graph is not meaningful to the readers, because no insights are given of DA among discriminant species (severity-driven P values ?). Please do a RF or VIP plot with variable importance. Or, at least draw P values for each species among all groups (Kruskal-Wallis) with post-hoc test (Mann-Whitney), but on transformed data!

R. Considering that variation of species among groups is analyzed in more detail by enhanced volcano, Random Forest and Lefseq, we decided to remove the initial analyses previously shown in figure 1C.

-line 270: I advise to have an insight on the 16S NCBI database, encompassing >50k genomes, updated every two-three days. In contrast, in eHOMD v3 (is that this version was used?) total number of genomes are 2087 including non-oral/non-nasal taxa. It means that some non-oral bacterial species should have been unrecognized or under-detected.

R. True, by including the NCBI database we could identify additional species, however, eHOMD has proved to be a comprehensive microbiome database for the study of microbiome in the aerodigestive tract with a better resolution than other common databases. Still, we acknowledge the limitation of our results and accept that it can be improved with the NCBI database, this is described in Discussion, lines 687-692.

-line 279: please report objective values of Delta among relative abundances or P values when dealing with comparisons, avoid subjective jargon such as "particularly higher" and so on.

R. We now indicate p-values instead of subjective phrases.

-line 296: why enhanced? In any case, Volcano plots do not visualize species with zero relative abundance in at least one group. I advise to draw a Figure1D panel reporting FoldC differences. Or, at least, a NMDS bi-plot, superimposing on Figure1B the eigenvalues of bacterial drivers, we should see on bottom-right corner species such as *A.baumannii*, *Prevotella* spp., *Veillonella* spp., etc..

R. Enhanced refers to enhanced- visual, where the X axis represents the FoldC.

-line 363: please cluster the bacterial species by the cohorts, reporting in descending order of "mean accuracy" and "LDA": in this manner it's easier for the Reader to follow results in a nutshell. I would also like to see a ROC curve with selected bacterial species able to separate 1vs1 the different cohorts.

R. Cohorts in Figure 3B were clustered and ordered as suggested and a ROC analyses with the LDA scores is now shown in the figure

-line 413-414: this is not the usual workflow for network co-abundance analysis. The best option is to do networks with all the species, then filtering edges with Pearson coefficients (e.g. 0.7) or P values (e.g. <0.01), then ruling out singletons, then assessing communities/clusters. In this way the network is unbiased for community and functional definition. I advise to calculate betweenness centrality and figure out keystone species in each cohort. I stress also the previous comment on drawing ROC curves using the selected bacterial species (here is right to use results from volcano, lfe, RF, and networks).

R. Analysis of network was done again using the suggested workflow and the paragraph of co-abundance modified accordingly (see lines 255-265) and results shown now in figure 5.

-line 419: transform this Table2 in an inset of Figure5 as stacked barplot. Erase then Table2.

R. Table 2 data is now presented as a figure 6.

-line 501: the Picrust2 dataset was normalized for total gene count before doing analyses? it seems to be a bias at level2, which affects also level3.

R. Details of the analysis of metabolic routes are described in lines 267-277

-line 561: see previous comment about eHOMD.

R= See our reply to the previous comment

-line 645: I would see also these network metrics computed for each severity-cohort and reported in Figure5: i) Average Path length; ii) density; iii) diameter; iv) number of communities.

R. Additional network metrics were measured and reported in figure 6.

April 17, 2023

Prof. Javier Torres
Mexican Institute of Social Security
Infectious Diseases
Av. Cuauthemoc 330
Av. Cuitlahuac # 2931 int 16
Mexico, D.F 92840
Mexico

Re: mSystems01062-22R1 (Diversity, composition, and networking of saliva microbiota distinguish the severity of COVID-19 episodes as revealed by an analysis of 16S rRNA variable V1-V3 regions sequences)

Dear Prof. Javier Torres:

Thank you to the authors for thoughtfully and thoroughly addressing the reviewer's comments. It is greatly appreciated

Your manuscript has been accepted, and I am forwarding it to the ASM Journals Department for publication. For your reference, ASM Journals' address is given below. Before it can be scheduled for publication, your manuscript will be checked by the mSystems production staff to make sure that all elements meet the technical requirements for publication. They will contact you if anything needs to be revised before copyediting and production can begin. Otherwise, you will be notified when your proofs are ready to be viewed.

If you would like to submit a potential Featured Image, please email a file and a short legend to msystems@asmusa.org. Please note that we can only consider images that (i) the authors created or own and (ii) have not been previously published. By submitting, you agree that the image can be used under the same terms as the published article. File requirements: square dimensions (4" x 4"), 300 dpi resolution, RGB colorspace, TIF file format.

We recognize that the video files can become quite large, and so to avoid quality loss ASM suggests sending the video file via <https://www.wetransfer.com/>. When you have a final version of the video and the still ready to share, please send it to mSystems staff at msystems@asmusa.org.

Sincerely,

Marc Cook
Editor, mSystems

Journals Department
E-mail: mSystems@asmusa.org